# OpenReview forum: "AutoRAS: Learning Robust Agentic Systems with Primitive Representations"
_ICML.cc/2026/Conference — ICML 2026 regular_

### Official Review · Reviewer_UTsE · 2026-03-13

**Soundness:** 2
**Presentation:** 2
**Significance:** 2
**Originality:** 2
**Overall Recommendation:** 3
**Confidence:** 4

**Summary:**

This work proposes a framework for automated designing robust agentic systems, with the key contribution being the generation of a sequence of symbolic primitives, including multiple safety primitives, such as prompt injection hygiene, trajectory auditing, cross-agent consensus, etc. On top of primitive sequence, it further allows optimization to jointly optimize for task correctness, safety signals, and costs, etc.

**Compliance With Llm Reviewing Policy:**

Affirmed.

**Final Justification:**

While I really appreciate the added SWE Bench Lite results and the added details, I guess some of my major concerns are not fully resolved, so I keep my original score.

The key concern is the overall pipeline is too complex and makes me feel incremental. Even the authors themselves acknowledge that using their framework to optimize a real-world SWE agent (which typically works for 10 minutes for each task instance) is **not feasible** due to the time constraint.

Note that real-world agents, such as SWE agent, CUA agent, and deep research agent, typically work for at least a few minutes for each task instance. This makes me further feel like the evaluated settings are too "toy" compared to real-world settings.

**Key Questions For Authors:**

1. Can authors provide more justification on why we need GFlowNet for optimization instead of simple LLM-based optimization?

**Limitations:**

I would suggest trying to apply this to a more realistic agent benchmark, such as the SWE bench and OSWorld.

**Strengths And Weaknesses:**

This work tackles an important problem, as agent robustness should be considered as part of the agent design and should be treated similarly to benign task correctness.
The proposed framework turns out to be effective in multiple settings, especially in terms of robustness, compared to state-of-the-art baselines.

The weakness is that the overall pipeline is too complex, while the studied setting is overly simplified. It is relatively easy and efficient to test your proposed agent framework when your task is to solve MMLU and MATH, but how about testing your agent framework on the SWE Bench? Then, an overly complicated optimization will be really painful since evaluating your framework on real GitHub instances will need a lot of time. I would prefer more description of the design of the primitive and how to instantiate them.

---

> ### Author Rebuttal · Authors · 2026-03-31
>
> # **Response to Reviewer UTsE**
>
> We thank the reviewer for the helpful feedback and positive assessment. We appreciate the reviewer’s recognition of the importance of robustness-aware agentic system design and the strong results. We address the main concerns below.
>
> > 1. On the current evaluation setting
>
> We respectfully believe the current setting is a reasonable and standard starting point for evaluating automated agentic system design, rather than an overly simplified one.
>
> 1)Our evaluation setting is largely **consistent with prior automated design work** such as **AFlow (ICLR 2025 Oral)** and **MaAS (ICML 2025 oral)**. It covers not only general QA/reasoning tasks (**MMLU, MSMARCO**) but also more challenging settings requiring multi-step reasoning or execution(**MATH**, **ProgramDev**).
>
> 2)**The baselines span three different categories:** single-model methods, fixed-topology multi-agent systems, and prior automated design methods. AutoRAS consistently outperforms all three while maintaining reasonable cost, as shown in **Sec. 5.4** and **Appendix F.6/F.7**.
>
> 3)**Our goal is to learn reusable design strategies.** The cross-dataset transfer results in **Table 14** show that policies learned on one dataset generalize to others, indicating AutoRAS captures workflow regularities beyond a single benchmark.
>
> > 2. Extension to more realistic benchmarks.
>
> We agree that more realistic agent benchmarks are important, and we therefore conducted an additional evaluation on **SWE-bench Lite**.
>
> 1）To examine scalability to more realistic tasks while controlling execution cost and evaluation time, we use **SWE-bench Lite** and follow the same **1:4 train/test split** used in our other benchmarks. We use **DeepSeek-V3.2** as the backbone model.
>
> 2）We compare AutoRAS against five baselines. The results are summarized below:
>
> |**Method**|**Vanilla**|**Attack**|**Cost($)**|
> |-|-|-|-|
> |DeepSeek-V3.2| 32.44|24.58| 1.3812|
> |AgentPrune|41.67|35.41|21.5861|
> |G-designer|33.33|27.71|15.3725|
> |AFlow|42.08|30.42|34.7248|
> |MaAS|36.67|23.95|5.7354|
> |AutoRAS|45.83|42.91|12.4156|
>
> 3)**Conclusions remain consistent.** AutoRAS achieves the **best performance before and after attack** with the **smallest degradation (2.92)**. Its cost is much lower than methods with comparable performance, while its robustness is substantially stronger than cheaper baselines.
>
> > 3. Primitive design and instantiation.
>
> We appreciate the reviewer’s request for more detail here, and this part can indeed be explained more clearly.
>
> 1)**The primitive space contains two major categories:** structural primitives and behavioral primitives, as introduced in **Sec. 3.2** and detailed in **Appendix B.1**. Structural primitives define workflow composition, such as sequential composition, parallel grouping, and branching. Behavioral primitives define node-level operations.
>
> 2)**Behavioral primitives further include both functional primitives and safety primitives.** Functional primitives include modules such as direct answering, CoT reasoning, refinement, ensembling, and programming. Safety primitives include filtering, hygiene, tool guarding, trajectory auditing, and cross-agent consensus. Their implementations are listed in **Appendix B.2**.
>
> 3)**Instantiation has two stages.** Each primitive has a base template defining its core role. The analyzer (**App. C.1**) then takes the query q and safety signals s to generate two supplementary guidance blocks (functional and safety), which are appended to the corresponding base prompts. This ensures primitives are executable units adapted to the current task and robustness state.
>
> 4)**From sequence to workflow.** As shown in **Fig. 2**, the stack-based compiler deterministically maps a primitive sequence into an executable workflow topology, using structural primitives to organize instantiated behavioral primitives into a complete system.
>
> > 4. Why GFlowNet instead of simple LLM-based optimization.
>
> We choose GFlowNet because the optimization target is a **large discrete space of legal workflows**, where it achieves better performance and robustness at lower cost than LLM-based search.
>
> 1)**Cost-efficiency.** **Cost-efficiency.** LLM-based search requires repeated LLM calls during optimization. In **Table 3**, **AFlow** incurs **$2.21** training cost on MMLU, while our policy requires only **$0.33**. On **SWE-bench Lite**, AFlow costs **$34.72** while AutoRAS costs **$12.42**.
>
> 2)**Equifinality challenge.** Many workflows can achieve similar utility and robustness. GFlowNet learns a **distribution over effective designs** rather than collapsing to a single candidate, naturally handling equifinality (**Sec. 1**, **Sec. 4.3**, **App. G.2**).
>
> 3)**Legality-constrained search.** The stack-based compiler mask (**Sec. 4.2**) ensures every generated sequence is executable, reducing the effective search space and stabilizing optimization.

---

> > ### Author Rebuttal · Reviewer_UTsE · 2026-04-03
> >
> > Thank authors for the response. The added SWE-Bench Lite results look quite interesting to me, could you provide more details on the derived robust agentic systems for SWE Bench, and how is it different from the agentic systems designed by baselines such as AFlow, and elaborate more about the baseline (are you using the default SWE agent scaffold, like mini swe agent?)

---

> > > ### Author Response · Authors · 2026-04-06
> > >
> > > Thank you again for the thoughtful follow-up and the additional questions. We are glad that the added **SWE-bench Lite** results help clarify the concern. Below, we provide more detail in two parts: **(1) the SWE-bench Lite setting**, and **(2) the derived systems and how they differ from the baselines**.
> > >
> > > **1. SWE-bench Lite setting.**
> > >
> > > 1）**Execution scaffold.** To balance evaluation fidelity with efficiency, we adopt a lightweight **three-stage** scaffold for SWE-bench Lite following the basic problem decomposition in [1]: **file localization → function localization → patch generation**, with one LLM call per stage and **DeepSeek-V3.2** as the backbone. We simplify the execution pipeline by using **single-generation** patch production and omitting multi-candidate sampling and test-time verification. In practice, open-ended mini-SWE-agent style ReAct loops often require **50+ interactions per issue**; in our setting, one such evaluation takes around **10 minutes on average**, and a full round of training and evaluation may repeat this process roughly **480 times**, making that setup impractical for rebuttal-time experimentation. At the same time, the Agentless-style scaffold still preserves the essential software-engineering structure, allowing the comparison to reflect differences in **agentic system design**, rather than merely scaffold choice.
> > >
> > > 2）**Initialization of design elements.** All baselines and our AutoRAS are built on top of the **same shared three-stage execution unit**. The difference is therefore not in giving AutoRAS a stronger underlying SWE solver, but in **how each method composes and invokes this shared unit**.
> > >
> > > - In our **AutoRAS**, primitives define the calling pattern: `Direct` executes the three-stage pipeline directly; `CoT` adds reasoning before or during the stages; `Refine` revises intermediate outputs; safety primitives insert filtering or verification before/after execution; and structural primitives determine how multiple such units are connected.
> > >
> > > - **AFlow** and **MaAS** use the same execution unit within their own operator/search spaces.
> > >
> > > - **AgentPrune** and **G-Designer** use the same scaffold as the node-level solver while optimizing communication or topology.
> > >
> > > **2. Derived systems and their differences from the baselines.**
> > >
> > > 1）**How the designed systems differ.** The main difference lies in the **level of search ability and robustness-awareness** in the resulting systems.
> > >
> > > - **AFlow** searches in a code-based space that is flexible but weakly constrained. In our SWE-bench Lite case study, it tends to produce systems that are mainly **performance-oriented**, such as candidate diversity, ensembling, and review.
> > > -  **AgentPrune / G-Designer** mainly optimize **communication topology** over a fixed set of agents. Their derived systems are therefore better understood as topology adjustments over predefined roles.
> > > - **MaAS** varies agent behaviors within a fixed supernet, but the structure itself is not fully searchable and robustness mechanisms are not explicitly part of the design space.
> > > -  **AutoRAS** jointly learns **how units are connected, what behaviors they execute, and where safety primitives should be inserted**. As a result, the derived SWE-bench Lite systems are not just more complex topologies; they are **robustness-aware systems**, with safety mechanisms inserted at stages where failures or attacks are most likely to propagate.
> > >
> > > 2）**Additional details and case studies.** We provide the concrete derived designs, execution cases, and implementation details at [here](https://anonymous.4open.science/r/AutoRAS_rebuttal-3857/swebench_details/Readme.md).
> > >
> > > **Reference**
> > >
> > > [1] *Agentless: Demystifying LLM-based Software Engineering Agents*.
> > >
> > > ---
> > >
> > > We sincerely thank the reviewer again for the thoughtful follow-up and for recognizing both the importance of incorporating robustness into agentic system design and the effectiveness of our framework. We hope this clarification helps further explain the SWE-bench Lite setting and the differences between the systems derived by our AutoRAS and the baselines.

---

### Official Review · Reviewer_GjrT · 2026-03-13

**Soundness:** 3
**Presentation:** 3
**Significance:** 3
**Originality:** 3
**Overall Recommendation:** 3
**Confidence:** 4

**Summary:**

This paper proposes AutoRAS, a framework for automatically designing robust multi-agent systems for LLM-based workflows by representing system architectures as sequences of symbolic primitives. The authors explore the concept of integrating structural and behavioral components of agentic workflows through a unified primitive representation, enabling automated search over system designs. This study aims to analyze a central concept of how robustness signals derived from execution traces can guide the optimization of agentic architectures. The framework combines primitive-sequence generation, robustness-aware execution, and flow-based learning to iteratively improve system design. Experiments across several reasoning and programming benchmarks suggest that the proposed approach improves both task performance and robustness against adversarial attacks compared to existing automated or manually designed agentic systems.

**Compliance With Llm Reviewing Policy:**

Affirmed.

**Final Justification:**

The paper presents a technically solid and well-motivated approach to robustness-aware automated agent design, with clear empirical improvements and a useful unified primitive formulation. The rebuttal clarified several implementation details and better articulated the intended contribution, particularly the joint modeling of structure, behavior, and safety.

However, my main concerns remain only partially addressed. The methodological novelty still appears incremental relative to prior work, the approach relies significantly on manually defined primitives and templates, and the evaluation does not fully establish broad robustness beyond the studied settings. As a result, while promising, the work would benefit from stronger differentiation and more comprehensive validation before being accepted.

**Key Questions For Authors:**

1. How does the proposed primitive representation fundamentally differ from the modular building blocks or workflow components used in prior automated agent design frameworks (AFlow, EvoAgentX, MaAS, etc.)?

2. To what extent does the system depend on manually defined primitive sets and prompt templates, and how sensitive are the results to these design choices (beyond just the count presented in Table 8)?

3. Can the authors clarify the training procedure and architecture used for the primitive generation policy, including whether the encoder–decoder model is trained from scratch or fine-tuned?

4. How comprehensive are the robustness signals derived from execution traces, and how well do they capture external adversarial behaviors beyond the specific attack scenarios evaluated?

5. The evaluation includes relatively small training datasets in some tasks. How stable are the results when scaling to larger datasets or more diverse task distributions?

**Limitations:**

Yes, concerning the "Requires vocabulary design" of Primitives.

**Strengths And Weaknesses:**

**Strengths**

(+) The paper addresses an important problem of incorporating robustness considerations directly into the automated design of agentic systems rather than treating robustness as a post-hoc defense mechanism.

(+) The formulation that represents agentic workflows as sequences of structural and behavioral primitives provides a clear abstraction that allows the design problem to be treated as a sequence-generation task.

(+) The experimental evaluation is relatively comprehensive across multiple tasks and adversarial settings, showing smaller degradation under attacks compared with several competitive methods.

(+) The paper includes additional analyses such as transferability across backbone models, sensitivity to primitive vocabularies, and cost analysis, which helps illustrate the broader behavior of the system.


---


**Weaknesses**

(-) The novelty of the primitive-based representation is somewhat unclear. The paper does not sufficiently clarify how the proposed primitives fundamentally differ from existing design elements used in earlier automated agent architecture frameworks.

(-) Several components appear to rely on manually designed elements, such as the primitive repository and base prompt templates, which raises questions about how much of the system design process is truly automated versus still human-dependent.

(-) The paper relies on execution-derived feedback and textual signals to guide optimization, but similar feedback-based approaches [a-c] have been explored in prior work, making it unclear what methodological innovation is introduced beyond combining existing ideas.

- [a] Chen, Xinyun, et al. "Teaching Large Language Models to Self-Debug." The Twelfth International Conference on Learning Representations, 2024.
- [b] Yao, Weiran, et al. "Retroformer: Retrospective Large Language Agents with Policy Gradient Optimization." The Twelfth International Conference on Learning Representations, 2024.
- [c] Trirat, Patara, Wonyong Jeong, and Sung Ju Hwang. "AutoML-Agent: A Multi-Agent LLM Framework for Full-Pipeline AutoML." International Conference on Machine Learning, 2025.

(-) The connection between robustness signals extracted from execution traces and the external robustness objectives is not fully clarified, making it difficult to assess whether the system effectively captures all relevant failure modes.

(-) The evaluation datasets appear relatively small in some cases, which may limit the reliability of the conclusions regarding generalization and robustness.

---
**Minor:** Typos "Exeution"? in Figure 2.

---

> ### Author Rebuttal · Authors · 2026-03-31
>
> ## **Response to Reviewer GjrT**
>
> We thank the reviewer for the thoughtful comments and respond point by point below.
>
> > W1&Q1: Difference between primitives and prior work?
>
> The key difference is that AutoRAS uses a **single searchable symbolic space** to jointly represent **topology, behavior, and safety**. We provide a more detailed comparison in **App. G**.
>
> **1）Unified representation.** Prior methods separate these aspects: graph-based methods optimize topology, operator-based methods vary behavior, and code-based methods are expressive but hard to optimize. Our primitive vocabulary contains **structural, functional, and safety primitives in one shared sequence space**.
>
> **2）Joint optimization of structure and behavior.** Since structural and behavioral primitives share one sequence space, the optimizer searches over **both structure and behavior simultaneously**. The discrete, legality-constrained space also makes optimization more tractable than code-based methods.
>
> **3）Safety is part of the design space itself.** Safety primitives are generated together with functional ones, so robustness is incorporated **during construction**, not as a post-hoc patch.
>
> > W2&Q2&L1: To what extent does the method depend on manually designed primitives and prompt templates?
>
> We thank the reviewer. We provide a brief answer here and refer the reviewer to **Reviewer KH4e (W1, W3)** for a fuller discussion. AutoRAS follows the same setting as prior automated design methods: predefined components specify the **design space**, not the final solution. The learned policy determines **which primitives to compose, in what order, and with what topology**.
>
> > W3: Is this more than combining existing feedback-based ideas?
>
> We thank the reviewer for pointing out these related works. We agree that feedback-based refinement is not new. However, prior methods use feedback to improve outputs or behaviors within a **fixed** system, whereas **AutoRAS uses execution feedback to optimize the system design itself, including composition, topology, and safety structure.** The contribution is feedback-driven robust system design, not feedback alone.
>
> > Q3: Training procedure and architecture of the primitive generation policy.
>
> The primitive generation policy is a lightweight encoder–decoder model,  separate from the backbone LLMs.
>
> **1)Architecture.** The policy has 4.29M parameters. Query $q$, safety signal $s$, and primitive descriptions are encoded with all-MiniLM-L6-v2. The encoder aligns them with the primitive space via cross-attention; the decoder scores candidates step by step via a bilinear projector. A compiler-derived legality mask prunes invalid choices online.
>
> **2)Training procedure.** For each query, the policy samples $K$=4 sequences. Each is compiled, executed, and scored by Eq. 8. Trajectories optimize the policy via the Trajectory Balance objective, which converges after ~30 trajectories (App. F.6).
>
> > W4&Q4: How comprehensive are the robustness signals, especially for external attacks?
>
> **1）Robustness signals.** The monitor inspects execution traces and produces both numerical scores and natural-language safety signals. For external robustness, it checks whether the sequence includes safety primitives and robust topologies. For internal robustness, it evaluates 9 failure modes following [1]. Scores $r_{\text{ext}}$, $r_{\text{int}}$ enter the reward (**Eq. 8**); textual signals s condition subsequent generation and refine primitives via textual gradients. The policy thus learns to generate designs with built-in defensive structures.
>
> **2）Generalization beyond evaluated attacks.**  Since $r_{\text{ext}}$ encourages general defensive structures rather than attack-specific defenses, the learned designs incorporate universal safeguards. **Table 14** shows that policies trained on one dataset transfer well to others and defend against different attacks, indicating the learned robustness patterns do not overfit to specific failure modes.
>
> [1]Why Do Multi-Agent LLM Systems Fail? (NeurIPS 2025).
>
> > W5&Q5:  Small training sets and stability.
>
> We agree that some training sets are relatively small, but this is due to the nature of the problem setting.
>
> **1）Optimization under limited training data.** Each query is sampled $K$=4 times, so even small datasets yield multiple training trajectories.
>
> **2）Empirical data efficiency.** **Tables 12,13** (**App. F.3**) show peak performance with few training queries (e.g., 15 on MSMARCO). The loss stabilizes after ~30 trajectories (**App. F.6**).
>
> **3）Generalization beyond the training set.** **Table 14** shows strong cross-dataset transfer across different formats and domains; e.g., a policy trained on **6 ProgramDev queries** generalizes to **486 MATH** test instances, suggesting AutoRAS learns reusable design principles rather than overfitting.
>
> > Minor Typos
>
> Thank you for the meticulous review! We will fix these typos in the revision.

---

> > ### Author Rebuttal · Reviewer_GjrT · 2026-04-03
> >
> > Thank you for the detailed rebuttal. The response helps clarify several implementation details, especially the primitive generation policy, training procedure, and the intended role of robustness signals. I also appreciate the added explanation of how the authors view the primitive space as jointly encoding topology, behavior, and safety.
> > However, my main concerns are only partially addressed.
> >
> > In particular, the distinction from prior automated agent design frameworks is clearer at a high level, but the methodological novelty still feels incremental relative to existing modular or feedback-driven design approaches. I also still have some concern about the degree of manual design involved in defining the primitive space and prompt templates, since these choices appear to substantially shape the search space itself. I further appreciate the clarification on robustness signals and transfer results, but I remain somewhat unconvinced that the current evaluation is sufficient to establish broad robustness beyond the studied attack settings, especially given the relatively small training sets in some tasks.
> >
> > Overall, the rebuttal improves clarity and addresses part of my confusion, but it does not fully change my original concerns about novelty, dependence on manually specified design components, and the scope of the robustness claims.

---

> > > ### Author Response · Authors · 2026-04-06
> > >
> > > We thank the reviewer again for the careful follow-up. We clarify the remaining concerns on **novelty**, **manual design dependence**, and **robustness generalization** below.
> > >
> > > ### 1. Methodological novelty beyond prior approaches
> > >
> > > To better clarify the contribution of AutoRAS, we summarize it in three aspects:
> > >
> > > 1）**A unified design representation.** System design is formulated as primitive-sequence generation in a single legality-constrained space that jointly encodes structure, behavior, and safety. As shown in **App. G / Table 16**, our AutoRAS can efficiently and jointly optimize topology, behavioral variation, and robustness-aware design.
> > >
> > > 2）**A robustness-aware design loop.** Rather than treating robustness as a post-hoc constraint, our AutoRAS incorporates execution-derived robustness signals directly into the design process, allowing the policy to learn from observed vulnerabilities and progressively refine system designs toward both accuracy and resilience.
> > >
> > > 3）**Flow-based trajectory-balance optimization.** The combinatorial design space exhibits equifinality that distinct workflows can achieve similar effectiveness. Trajectory Balance over GFlowNet provides a principled way to explore this space, avoiding single-solution collapse by distributing probability mass across diverse, high-quality designs.
> > >
> > > ### 2. Dependence on manually specified primitives and prompt templates
> > >
> > > We agree that the primitive space and base templates shape the search space, but this is inherent to automated design: one must choose a representation before learning over it. The key question is whether the space is stable and effective, and whether the learned policy contributes beyond it.
> > >
> > > 1）**Some basic manual design is necessary in automated design problems to define the search space.** The optimizer must operate over a concrete and executable set of design elements. In our AutoRAS, the primitive vocabulary and base templates define the available building blocks (App. G), while the learned policy determines which primitives to compose, in what order, and with what topology.
> > >
> > > 2）**The current vocabulary is well balanced.** **Table 8** shows that removing primitive categories reduces performance, which tests the necessity of each component, not a scenario where primitives would be removed in practice. Adding more primitives yields only marginal gains, confirming the current set is balanced rather than fragile.
> > >
> > > 3）**How our AutoRAS learns within this space also matters.** Our goal is not to enumerate all possible workflow patterns, but to discover effective and robust ones through distributional exploration over a representative primitive set. This is where **equifinality** (**Sec. 1, Challenge (iii)**) becomes central. Even under the same primitive space, replacing GFlowNet with **PPO / REINFORCE / MCTS** leads to consistently worse results (see **Reviewer KRuS, W2**), showing that gains come from the representation–learning combination, not the vocabulary alone.
> > >
> > > ### 3. Scope and generalization of the robustness claims
> > >
> > >
> > > We agree that establishing broad robustness is a central challenge in automated agentic system design. As discussed in **Sec. 1 (Challenge ii: Unpredictability)**, the open-ended nature of attacks makes them difficult to anticipate. Our claim is that our AutoRAS learns **general defensive structures**, rather than attack-specific patches.
> > >
> > > 1）**Coverage of the evaluated attacks.** In **App. A.3**, we evaluate four attack types targeting different layers of an agentic system: **brain attack** (prompt layer), **memory attack** (knowledge layer), **tool attack** (tool-use layer), and **agent-to-agent attack** (communication layer). Across four datasets with different attack combinations, these cover the major attack surfaces of agentic architectures.
> > >
> > > 2）**Existing evidence for defense generalization.** In **Table 14**, different datasets use different attack combinations  (e.g., **MMLU: brain + agent-to-agent; MSMARCO: brain + memory + agent-to-agent**), yet policies trained on one dataset transfer effectively to others, suggesting the learned patterns are not tied to one attack configuration.
> > >
> > > 3）**Additional evaluation on stronger unseen attack variants.** We tested the trained AutoRAS policy against stronger variants from [1] and [2], both unseen during training. These correspond to stronger **brain-attack** and **agent-to-agent** variants. The results are:
> > >
> > > |Method|MMLU Attack|Cost(\$)|MSMARCO Attack|Cost(\$)|
> > > |-|-|-|-|-|
> > > |AgentHarm variant|81.17|0.6614|86.25|0.4958|
> > > |ASB variant|81.70|0.6583|85.00|0.4879|
> > > |AutoRAS|82.35|0.6655|88.75|0.5066|
> > >
> > >
> > > These results suggest that our AutoRAS learns general defensive structures rather than overfitting to the original attack prompts.
> > >
> > > [1] AgentHarm: A Benchmark for Measuring Harmfulness of LLM Agents. ICLR 2025.
> > >
> > > [2] Agent Security Bench (ASB): Formalizing and Benchmarking Attacks and Defenses in LLM-based Agents. ICLR 2025.

---

### Official Review · Reviewer_KH4e · 2026-03-13

**Soundness:** 3
**Presentation:** 1
**Significance:** 3
**Originality:** 3
**Overall Recommendation:** 4
**Confidence:** 2

**Summary:**

The paper proposes **AutoRAS**, a framework for automatically designing robust agentic systems by representing a system as a sequence of symbolic primitives that jointly encode both structure and behavior. The workflow is as follows. A generator first produces a primitive sequence that specifies the system design. This sequence is compiled into an executable agentic system under a set of legality constraints. The system is then executed and its trace is monitored for task performance, cost, and robustness failures. These signals are used to optimize the generator using a GFlowNet-style trajectory balance objective together with textual feedback that edits primitive prompts. The central claim is that robustness should be integrated into the design loop itself rather than applied as a post hoc patch, and that primitive representations provide a tractable search space for this optimization.

Conceptually, the paper introduces three main components. First, it defines an agentic system more broadly than a directed acyclic graph by incorporating behavioral policies, aggregation mechanisms, and interaction round semantics. Second, it introduces a primitive vocabulary that encodes both structural choices and behavioral decisions within a single symbolic sequence. Third, it uses execution-derived signals, including external robustness signals and internal reliability metrics, to guide sequence optimization through flow-based learning.

AutoRAS is evaluated on **MMLU, MSMARCO, MATH, and ProgramDev** under both standard and adversarial settings. The method is compared against single-agent baselines, manually designed multi-agent systems, and prior automatic workflow design approaches. The main empirical result is that AutoRAS achieves the strongest average performance across tasks and exhibits the smallest degradation under adversarial conditions in the primary evaluation table. The paper also reports transfer across multiple backbone models, an efficiency and cost analysis, sensitivity studies, and ablations indicating that robustness signals and explicit robustness objectives play an important role in the observed gains.

**Compliance With Llm Reviewing Policy:**

Affirmed.

**Key Questions For Authors:**

1. **Learning vs. predefined components.**
   How much of AutoRAS is actually learned versus predefined? In particular, how dependent are the results on hand-designed primitive templates, the analyzer that generates supplementary prompts, and the monitoring LLM?

2. **Textual gradient mechanism.**
   How exactly are textual gradients converted into prompt edits for primitive sequences? The paper mentions this step but does not clearly specify the algorithmic process.

3. **Dependence on monitor and analyzer models.**
   Since robustness and reliability rewards rely on monitor-flagged failures, how sensitive are the results to the choice of monitor/analyzer models and prompts?

4. **Reward formulation.**
   The robustness rewards ((1-p)^m) and ((1-p)^k) assume equal severity and independence of failures. Did the authors explore alternative formulations that weight different failure types?

5. **Cost and scalability.**
   How does the search cost scale with larger primitive vocabularies, longer sequences, or more complex tasks, and in what regimes does the added design cost justify the robustness gains?

**Limitations:**

Not much captured but also not much needs to be addressed on the societal impact for the paper.

**Strengths And Weaknesses:**

**Strengths**

1. The paper has a clear thesis. Unlike other agent papers that treat robustness as an auxiliary runtime defense problem, this one argues that robustness should shape the actual system design space and optimization objective.

2. The primitive representation is a reasonable contribution. It sits in a middle ground between graph-only, operator-only, and code-based representations. The paper correctly identifies the key tradeoff: graphs express topology but not behavioral detail, code is expressive but difficult to constrain and optimize, and operators lack explicit structural semantics. The primitive representation is presented as a compromise across these options.

3. The paper explicitly tracks correctness, cost, external robustness, and internal reliability, and incorporates these signals into reward shaping. The ablation table supports the claim that removing the robustness signal materially hurts attack performance.

4. Empirically, the main table is strong. The paper’s average vanilla score is the best among the compared methods, but the more convincing result is the attack robustness. The reported average drop is 2.13%, compared with substantially larger drops for the other automatic workflow designers.

5. The sensitivity and ablation sections are useful. The paper identifies saturation around sequence length 16 and sampling count 4 for most settings, and the ablations isolate the importance of the safety signal, text gradient, and robustness reward terms.

---

**Weaknesses**

1. The biggest issue is that the paper’s novelty is somewhat overstated relative to what is actually implemented. The primitive sequence formulation is useful, but much of the actual behavior still comes from manually designed primitive templates, an analyzer LLM that writes supplementary prompt blocks, and an LLM-based monitor that audits failures. The framework is therefore not truly end-to-end learning of robust agentic systems, but rather a structured search over hand-defined symbolic operators with LLM assistance layered on top.

2. The methodology section is under-specified where it matters most. The paper presents appealing equations for trajectory balance and reward shaping, but the engineering details that determine whether the method meaningfully improves over prompted search are not sufficiently explained in the main text. For example, how textual gradients are converted into prompt edits is vague, and the prompting details for the analyzer and monitor are largely deferred.

3. The reward design is simplistic. External robustness is computed as ((1-p)^m) and internal reliability as ((1-p)^k), where (m) and (k) are counts of violations and failures flagged by the monitor. This formulation assumes that all failure events of a given type have equal severity and independent effects.

4. The cost story is mixed. The paper frames AutoRAS as cost-efficient and second lowest in total cost among the compared automatic systems on MMLU, and Table 3 does support that narrow claim. However, the broader claim of favorable cost tradeoffs is less convincing. AutoRAS remains meaningfully more expensive than MaAS, and its relative advantage partly reflects the fact that MaAS collapses under attack in this setup.

5. Writing quality is uneven. There are noticeable grammar errors, formatting inconsistencies, and awkward phrasing throughout the paper, which makes parts of the text difficult to follow.

---

> ### Author Rebuttal · Authors · 2026-03-31
>
> ## **Response to Reviewer KH4e**
>
> We thank the reviewer for the detailed assessment and address each concern below.
>
> > W1&Q1. What is learned in AutoRAS?
>
> We would like to clarify that AutoRAS addresses the same problem setting as prior work such as AFlow (ICLR 2025 oral) and MaAS (ICML 2025 oral): learning how to compose predefined behavioral building blocks into stronger agentic systems.(see **App. G**)
>
> **1)What is predefined.** The primitives vocabulary, the base prompt for each primitive, and the prompts used by the analyzer and monitor modules. This is analogous to defining an action space in RL: it specifies the available building blocks, but not the final solution.
>
> **2）What is learned.** The **design policy**, which decides **which primitives to compose, in what order, and with what topology**. Safety signals from execution are fed back to guide future designs, and the supplementary prompt blocks are refined accordingly.
>
> **3）Empirical evidence.** The gains mainly come from learned design rather than static templates. In **Table 4**, removing the safety signal causes the largest attacked-performance drop, from **82.35 to 71.90**. We also replaced GFlowNet with **PPO, REINFORCE, and MCTS** under the same primitive space, our method remains best in both accuracy and robustness at comparable cost (see **Reviewer KRuS W2**).
>
> > W2&Q2. Further clarification of textual gradients and the other components.
>
> We briefly clarify the main workflow.
>
> **1)Sequence generation.** The policy network takes the query $q$, previous safety signals $s$ (empty at initialization), and primitive descriptions as input, then generates a primitive sequence, which is compiled and executed.
>
> **2)Trace analysis.** Execution produces a structured trace $\mathcal{T}$. The monitor (**App. E.1**) analyzes $\mathcal{T}$, outputs safety signals s and robustness scores $r_{\text{ext}}$, $r_{\text{int}}$ (details in **Reviewer GjrT W4**), which enter **Eq. 8** to update the policy via the Trajectory Balance loss.
>
> **3)Prompt refinement via textual gradients.** The analyzer (**App. C.1**) takes q and s and generates two guidance blocks(functional and safety), which are merged into a prompt library and appended to corresponding base prompts. The updated s also conditions the next round of sequence generation.
>
> > W3&Q4. Reward formulation
>
> We agree that the current reward is deliberately simple, but it is effective in practice.
>
> **1)Design rationale.** Any detected failure indicates that the system needs improvement. We therefore prioritize **failure occurrence and accumulation** over modeling detailed severity or dependency patterns. The $(1-p)^m$ and $(1-p)^k$ terms provide a simple monotonic penalty that is stable and easy to optimize.
>
> **2)Alternative formulations.**  We compared our default reward against two alternatives on **MMLU** and **MATH**:
>
> - **Severity-weighted**: assigns different weights to failure types (relaxing equal-severity);
> - **Compounding penalty**: replaces $(1-p)^m$ with $\exp(-\lambda m^2)$ (relaxing independence).
>
> |System|MMLU|||Math|||
> |-|-|-|-|-|-|-|
> ||Vanilla|Attack|cost($)|Vanilla|Attack|cost($)|
> |Severity-weighted|83.66|82.35|0.6671|57.41|55.34|6.2748|
> |Compounding penalty|82.35|81.17|0.6606|57.20|54.52|6.3744|
> |AutoRAS|83.01|82.35|0.6655|57.41|54.94|6.3453|
>
> 3）**Discussion.** All three variants perform similarly, suggesting our simplified reward is already effective and more complex designs do not yield clear gains.
>
> > W4&Q5. Cost story and scalability behave across different regimes, and when is the added cost justified?
>
> We agree that the key perspective is the **cost–robustness tradeoff**, rather than the cheapest absolute cost.
>
> **1)Cost-efficient robustness.** As shown in **Table 3**, AutoRAS achieves the **best performance** on MMLU with the **second-lowest total cost**. Its attacked drops is only **0.66**, while the cheaper **MaAS** drops by **15.16**. Lower cost does not imply better utility when robustness matters.
>
> **2)Scalability with larger design spaces.** The search cost scales reasonably as the design space grows.
>
> - In **Table 8**, expanding the number of primitives from **9 to 18** increases cost from **\\$0.42 to \\$0.72** with steady performance gains.
>
> - in **Fig. 3(a)** increasing the sequence length $L$ from **8 to 24** raises cost moderately, from about **\\$0.67 to \\$0.83**, with performance already saturating around $L$=16.
>
>
> **3)Scalability to more complex tasks.** We additionally evaluate on **SWE-bench Lite** under a more complex task setting, detailed results in our response to **Reviewer UTsE**. The conclusions remain consistent.
>
> > Q3. Dependence on analyzer and monitor models.
>
> We analyze this in **Appendix C.2** and **E.2**. Specifically, we compare 4 analyzer backbone models and 5 monitor implementations, and report their impact on both AutoRAS performance and cost.
>
> > W5. Writing quality
>
> We will carefully polish grammar, notation, and phrasing in the revision.

---

> > ### Author Rebuttal · Reviewer_KH4e · 2026-04-05
> >
> > Thank you for the detailed and well-structured rebuttal. The clarifications regarding the division between learned components and predefined primitives, as well as the additional details on the textual gradient mechanism and reward design, are helpful in better understanding the system. The additional experimental evidence on alternative reward formulations and scalability is also appreciated. That said, my primary concerns around the extent of end-to-end learning, reliance on auxiliary LLM components, and under-specification of key implementation details in the main paper remain only partially addressed. As such, I will keep my original score unchanged.

---

> > > ### Author Response · Authors · 2026-04-07
> > >
> > > We thank the reviewer again for the detailed follow-up and for acknowledging that the clarifications and additional experiments improved the understanding of the system. We understand that the remaining concerns are mainly about methodology detail and presentation granularity, and we address them briefly below.
> > >
> > > ### 1. On the extent of end-to-end learning
> > >
> > > AutoRAS learns the system-level design policy end to end over a concrete executable design space. Specifically, the process can be understood as follows.
> > >
> > > 1）**The learned policy determines the final design.** The primitive vocabulary and base templates define the available building blocks, analogous to an action space in RL. The learned policy decides **which primitives to compose, in what order, and with what topology**, conditioned on execution-derived robustness signals. Safety signals from execution are fed back to guide future designs, and supplementary prompt blocks are refined accordingly through textual feedback.
> > >
> > > 2）**The gains come primarily from the learned policy.** In **Table 4**, removing the safety signal causes the largest attacked-performance drop, showing that feedback-driven learning is the most critical component. Under the same primitive space, replacing GFlowNet with **PPO, REINFORCE, or MCTS** yields consistently worse accuracy and robustness at comparable cost (see our response to **Reviewer KRuS, W2**). This confirms that the performance gains are not explained by predefined components alone; both the design representation and the learning algorithm matter.
> > >
> > > ### 2. On reliance on auxiliary LLM components
> > >
> > > The analyzer and monitor are auxiliary modules whose current instantiations already provide a strong balance of reliability, effectiveness, and efficiency, while the framework remains compatible with stronger future replacements.
> > >
> > > 1）**Analyzer.** As shown in **Table 9 (App. C.2)**, stronger analyzer backbones yield only marginal improvements while incurring much higher cost. GPT-5 attains the best vanilla accuracy, but at over **400×** the cost of GPT-4o-mini. Moreover, **Table 4** shows that removing textual gradients has the smallest impact among the major ablations, indicating that the analyzer is helpful but not the main source of the system’s gains.
> > >
> > > 2）**Monitor.** As shown in **Table 10 (App. E.2)**, we compare **5 monitoring strategies**, including three adapted from prior work, and our default monitor provides the best balance of robustness, accuracy, and cost. We also provide additional monitor validation in our follow-up response to **Reviewer KRuS**, including human-annotated accuracy and random-monitor ablations. Together, these results suggest that the monitor is both practically reliable and not overly brittle to implementation choice.
> > >
> > > ### 3. On implementation details in the main paper
> > >
> > > We further clarify the relevant implementation details here to make the overall method easier to follow.
> > >
> > > 1）**Primitive space and instantiation.** The primitive taxonomy is defined in **App. B.1**, with concrete implementations in **App. B.2**. The two-stage instantiation process (base template + analyzer-generated supplementary block) is described in **Sec. 4.1** and detailed in **App. C.1**. The sensitivity analysis in **App. B.3 / Table 8** further validates that the vocabulary is well balanced.
> > >
> > > 2）**Monitor and robustness signals.** The 9-class failure taxonomy is fully specified in **App. E.1**, and monitor transferability across implementations is evaluated in **App. E.2 / Table 10**. Inter-judge agreement is reported in **App. A.1**.
> > >
> > > 3）**Execution and compilation.** The stack-based compiler is described in **Sec. 4.2**, with a complete execution example in **App. D**. The training behavior of the policy is analyzed in **App. F.6**.
> > >
> > > ### A broader view of the contribution
> > >
> > > We hope these clarifications help clarify the broader contribution of AutoRAS to automated agentic system design. To the best of our knowledge, AutoRAS is the first framework that **(1)** unifies topology, behavior, and safety in a single searchable primitive representation, **(2)** integrates robustness directly into the design loop through execution-derived signals, and **(3)** uses flow-based trajectory-level optimization to handle equifinality in the combinatorial design space. Empirically, AutoRAS achieves the best average performance across four benchmarks in both vanilla and adversarial settings, with only 2.13% average degradation under attack, while also transferring across datasets and backbone models at competitive cost. We believe these contributions, in both formulation and empirical validation, represent a meaningful advance, and we hope the reviewer will take them into account in the final assessment.

---

### Official Review · Reviewer_KRuS · 2026-03-13

**Soundness:** 1
**Presentation:** 1
**Significance:** 2
**Originality:** 2
**Overall Recommendation:** 4
**Confidence:** 3

**Summary:**

This paper focuses on the problem of automatically designing multi-agent systems that are robust under adversarial settings. To address this, the authors propose AutoRAS, a framework that represents multi-agent workflows as a sequence of symbolic primitives. A generated primitive sequence is compiled into a multi-agent system, executed on tasks, and evaluated using metrics including task accuracy, robustness to adversarial attacks, and execution cost. The primitive generation policy is then optimized using a GFlowNet-based objective to sample high-reward workflows. Experiments across reasoning, QA, math, and coding benchmarks show improved robustness under simulated adversarial attacks compared to prior automated agent design methods.

**Compliance With Llm Reviewing Policy:**

Affirmed.

**Final Justification:**

This paper studies an interesting problem of automatically constructing multi-agent systems that are robust to malicious attacks. While there were some issues, the additional experiments provided during the rebuttal have fully addressed them. The only remaining concern is presentation: the paper contains several typos and imprecise notations, which the authors have acknowledged. This is something the authors can address in the final revision. Given these improvements, I am updating my recommendation from weak reject to weak accept.

**Key Questions For Authors:**

- Are there any advantages of GFlowNets over standard policy-gradient RL methods (e.g., PPO, REINFORCE) or MCTS for optimizing the primitive sequence policy? Could the authors provide such a comparison?
- How are the primitives chosen?

**Limitations:**

yes

**Strengths And Weaknesses:**

## Strengths
- **Interesting and important problem**: The paper studies the problem of automatically generating multi-agent systems that are robust to attacks. This is a timely and important direction as agentic workflows become increasingly common. To the best of my knowledge, relatively few prior works have explicitly studied robustness in the context of automated multi-agent system design.
- **Extensive experimental evaluation**: The paper presents experiments across multiple benchmarks and adversarial settings. The results consistently show that the proposed method improves both task performance and robustness of the generated multi-agent systems compared to several baselines.

## Weaknesses
- **Robustness metric is weak**: The paper measures robustness through two metrics:
  - **External robustness**: resistance to adversarial attacks, $r_{ext} = (1-p)^m$
  - **Internal reliability**: resistance to system failures, $r_{int} = (1-p)^k$
  - where $m$ and $k$ denote the numbers of violations detected by an LLM judge while monitoring the execution trace. This definition depends on whether LLMs can reliably detect failures. However, existing literature has shown that LLM-based evaluation can be unreliable [1].
- **Limited ablation on the optimization method**: The paper does not explain why GFlowNets are chosen as the optimization method. Are there any advantages of GFlowNets over standard policy-gradient RL methods (e.g., PPO, REINFORCE) or MCTS for optimizing the primitive sequence policy? Could the authors provide such a comparison?
- **Primitive vocabulary design is somewhat arbitrary**: The performance of the multi-agent system heavily depends on the predefined set of premitives. However, the paper does not discuss how the primitives are chosen.
- **Poor clarify in the preliminary and methodology sections**: Several core concepts and notations are introduced without sufficient explanation or appear inconsistent. For example, on L161 it is unclear what $\Phi^*$ represents. The notation $\mathcal{S}(x)$ is also confusing, does it refer to a system generated from a sequence (x), and if so, should the notation instead be $\mathcal{S}(\mathcal{X})$ where $\mathcal{X}$ denotes a sequence of primitives? In Equation (3), the argmax on the right-hand side appears to be taken over elements of the primitive alphabet, while the left-hand side represents a sequence of primitives, which seems inconsistent. Additionally, on L238 the function $\mathcal{F}$ is used without being defined anywhere in the paper.

**References**
- [1] Which Agent Causes Task Failures and When? On Automated Failure Attribution of LLM Multi-Agent Systems. ICML 2025.

---

> ### Author Rebuttal · Authors · 2026-03-31
>
> ## **Response to Reviewer KRuS**
>
> We thank the reviewer for the careful reading and constructive feedback, and respond point by point below.
>
> > W1. Robustness metric and the use of LLM-based judges.
>
> We agree that LLM-based failure detection can be unreliable at very fine granularity, but our monitor operates at a **coarser and easier level** than the agent-/step-level attribution studied in [1], making it more reliable in our setting.
>
> **1）Coarser failure detection than [1].** Our monitor does not identify the exact agent or step causing a failure. Instead, it classifies the full execution trace into **9 trajectory-level failure types** following [2]. This is easier than the fine-grained attribution in [1]. Definitions are in **App. E.1**.
>
> **2）Validated and replaceable monitor.** We compare **5 monitoring strategies** in **Table 10**, including three adapted from [1]. We also report **Cohen's** $\kappa$ = 0.86 between cross-model LLM judges in **App. A.1**, indicating strong agreement. The monitor is a **plug-in module**, so stronger future attribution methods can be directly incorporated.
>
> **3）Effective even without perfect detection.** Optimization aggregates signals over many trajectories, so occasional misclassification does not dominate. Trajectory Balance learns a **distribution over designs**, more robust to noisy rewards than greedy optimization. In **Table 1**, AutoRAS shows only **2.13% average degradation** under attack.
>
> [1]Which Agent Causes Task Failures and When? ...(ICML 2025).
>
> [2]Why Do Multi-Agent LLM Systems Fail? (NeurIPS 2025).
>
> > W2&Q1. Why we use GFlowNets.
>
> 1）**Theoretical motivation.** The theoretical rationale is discussed in our response to **Reviewer UTsE (Point 4)**.
>
> 2）**Empirical comparison.** We compared against alternative optimizers on **MMLU** and **MATH**. The results are shown below:
>
> |Method|MMLU|||MATH|||
> |-|-|-|-|-|-|-|
> ||Vanilla|Attack|cost($)|Vanilla|Attack|cost($)|
> |PPO|79.08|73.20|0.6739|52.88|49.79|6.2745|
> |REINFORCE|78.43|75.82|0.6706|53.29|49.79|6.3415|
> |MCTS|81.05|79.74|1.6428|54.11|51.44|15.7671|
> |AutoRAS|83.01|82.35|0.6655|57.41|54.94|6.3453|
>
> **Result.** GFlowNet achieves the best vanilla and attacked performance with the smallest degradation on both datasets, at cost comparable to PPO/REINFORCE and far below MCTS.
>
> > W3&Q2. How the primitive vocabulary is chosen?
>
> The primitive vocabulary is not chosen arbitrarily; it is designed according to **three principles: grounding in prior literature, coverage of core workflow design dimensions, and empirical validation**.
>
> **1)Grounded in established components from prior work.** Functional primitives such as CoT, refinement, and ensembling correspond to widely used LLM reasoning strategies. Safety primitives such as filtering, hygiene, trajectory auditing, and cross-agent consensus are motivated by common failure types and attack patterns discussed in prior studies. Structural primitives such as SEQ, PAR, and FORK capture canonical workflow composition patterns.
>
> **2)Covering three core aspects of robust agentic system design: structure, task-solving behavior, and safety mechanisms.** This is why the vocabulary contains structural primitives, functional behavioral primitives, and safety behavioral primitives in one shared space.
>
> **3)Empirically validated rather than assumed.** In **Appendix B.3 / Table 8**, removing important primitive categories degrades performance, while adding extra primitives yields only marginal gains. This suggests that the current vocabulary is reasonably balanced rather than arbitrary, over-specified, or under-specified. The framework is also extensible: new primitives can be added without changing the overall method.
>
> > W4. On clarity issues in the preliminary and methodology sections.
>
> We thank the reviewer for pointing out these notation and presentation issues. These are valid concerns, and the intended meanings are as follows.
>
> 1)**On** $\Phi^*$: this denotes the space of primitive sequences formed from the primitive alphabet $\Phi$, rather than a single primitive. At each generation step, the actual selectable set is the legal subset determined by the compiler state.
>
> 2)**On** $S(x)$ **vs.** $S(\mathcal{X})$: the reviewer is correct that this notation is confusing. The intended meaning is that $X = (x_1,\dots,x_L)$ denotes a primitive sequence, and $S(\mathcal{X})$ denotes the compiled system generated from that sequence. We will use $S(\mathcal{X})$ consistently to distinguish sequences from individual primitives.
>
> 3)**On Eq. (3)**: the objective should be written over **legal primitive sequences**, not over single primitives. Each $x_i$ is selected step by step to form a complete legal sequence $\mathcal{X}$, and the optimization target is maximizes $R(S(\mathcal{X}))$.
>
> 4)**Undefined function near L238.** The function near L238 refers to the **set of feasible primitives at state** $z_{t-1}$, determined by the stack-based compiler. We will define it explicitly.

---

> > ### Author Rebuttal · Reviewer_KRuS · 2026-04-02
> >
> > > **W1. Robustness metric and the use of LLM-based judges.**
> >
> > 1. Could the authors provide an evaluation of the accuracy of the monitoring component? For example, it would be helpful to manually annotate a small set of trajectories and then evaluate failure detection accuracy against this ground truth.
> > 2. Additionally, I am curious about the worst-case scenario: if the monitoring component were replaced with random guessing, how significantly would that impact the final performance of the method?
> >
> > > **W2 & Q1. Why we use GFlowNets.**
> >
> > Thank you for the comparison with alternative optimization approaches. The results suggest that GFlowNets provide clear advantages and are not just chosen arbitrarily. This concern is fully resolved.
> >
> > > **W3 & Q2. How the primitive vocabulary is chosen.**
> >
> > Appendix B.3 provides a thorough sensitivity analysis of the primitive vocabulary, which I missed in my initial review. Thank you for pointing this out. This concern is fully resolved.
> >
> > > **W4. On clarity issues in the preliminary and methodology sections.**
> >
> > Thank you for acknowledging this concern and providing clarifications. I encourage the authors to address these presentation issues in the revision, as improving clarity would strengthen the credibility and professionalism of the paper.
> >
> > **Additional questions**
> >
> > I noticed that the coefficients $\alpha, \rho, \eta$, and $\beta$ in Equation 8 are not clearly specified. In the implementation details, the authors state:
> >
> > >"We set the maximum sequence length to $L = 16$, the cost parameter to $c = 0.2$, both external and internal robustness coefficients to $r_\text{ext} = 0.1, r_\text{int} = 0.1$, and the number of training samples per iteration to $k = 4$.
> >
> > It is unclear how these relate to the coefficients in Equation 8. Did the authors intend to say they set $\rho = 0.1, \eta = 0.1$, and $\beta = 0.2$? My understanding is that $c, r_{\text{ext}}$, and $r_{\text{int}}$ are runtime quantities computed from execution, rather than fixed hyperparameters?

---

> > > ### Author Response · Authors · 2026-04-06
> > >
> > > We thank the reviewer for the thoughtful follow-up questions. We are glad that W2/Q1 and W3/Q2 are resolved. Below, we provide further clarifications on the reliability of the monitor and the notation in Eq. 8.
> > >
> > > > **W1. Robustness metric and the use of LLM-based judges**
> > >
> > > **1. Monitor accuracy evaluation**
> > >
> > > We agree that directly evaluating the accuracy of the monitor against human annotation is very important. We therefore manually annotated **100 execution trajectories** from our experiments (**MMLU** and **MATH**), covering the 9 failure types used by our monitor. Since each trajectory is labeled for all 9 types, this yields **900 binary judgments** in total. We then evaluated several LLM backbones as monitors against this human-annotated ground truth and report label-level accuracy below.
> > >
> > > | Monitor Backbone     | Accuracy | Cost (\$) |
> > > | -------------------- | -------: | -------: |
> > > | GPT-4o-mini (ours)   |   0.9489 |     0.31 |
> > > | GPT-5.4              |   0.9644 |     4.89 |
> > > | Gemini-3-pro-preview |   0.9656 |     3.62 |
> > > | DeepSeek-V3.2        |   0.9622 |     0.17 |
> > >
> > > These results complement the previously reported **cross-model Cohen’s $\kappa$ = 0.86** and suggest that **trajectory-level failure typing is a tractable task for current LLM monitors**, even though finer-grained step-/agent-level attribution remains harder. The annotated data and monitor outputs are available [here](https://anonymous.4open.science/r/AutoRAS_rebuttal-3857/manually_annotate_exp/README.md).
> > >
> > > **2. Random monitor ablation**
> > >
> > > We also tested the case suggested by the reviewer by replacing the monitor with **random prediction**. Concretely, each failure type is marked as true with a fixed random probability (11%), and the textual safety signal is replaced by a fixed template corresponding to the sampled failure type. All other components remain unchanged.
> > >
> > > | Method         | MMLU Vanilla | MMLU Attack | Cost (\$) | MATH Vanilla | MATH Attack | Cost (\$) |
> > > | -------------- | -----------: | ----------: | -------: | -----------: | ----------: | -------: |
> > > | Random monitor |        81.17 |       79.74 |   0.6571 |        55.14 |       53.50 |   6.3512 |
> > > | AutoRAS    |    83.01 |   82.35 | 0.6655 |    57.41 |   54.94 | 6.3453 |
> > >
> > > This ablation shows two things:
> > >
> > > **(i)** Replacing the monitor with random guessing does reduce performance in both vanilla and attacked settings, confirming that a well-designed monitor provides more informative optimization signals.
> > >
> > >  **(ii)** The degradation is not catastrophic, which is consistent with our earlier point that optimization aggregates signals over many trajectories rather than relying on any single judgment. Overall, this supports the claim that the monitor is useful, while the framework is not brittle to occasional monitoring noise.
> > >
> > > > **Additional question: coefficients in Eq. 8**
> > >
> > > We appreciate the reviewer for pointing out this typo.  We correct the expression of Eq.8 as follows.
> > >
> > > In **Eq. 8**, $R(X) = \alpha u + \rho r_{\text{ext}} + \eta r_{\text{int}} - \beta c,$ the quantities **$\(u, r_{\text{ext}}, r_{\text{int}}, c\)$** are **runtime values** computed from execution and normalized to \([0,1]\), while **$\(\alpha, \rho, \eta, \beta\)$** are **fixed trade-off hyperparameters**. Our implementation uses $\alpha = 1$ (implicit), $\rho = 0.1$, $\eta = 0.1$, and $\beta = 0.2$. Thus, the sentence in the implementation details near **R296** saying  "$\(c = 0.2, r_{\text{ext}} = 0.1, r_{\text{int}} = 0.1\)$" is a typo: these were intended to refer to the **coefficients** $\(\beta, \rho, \eta\)$, not the runtime scores themselves.
> > >
> > > For completeness, we also tested the sensitivity of $\beta$:
> > >
> > > | \(\beta\) |    0.1 |    0.2 |    0.3 |    0.4 |    0.5 |
> > > | --------- | -----: | -----: | -----: | -----: | -----: |
> > > | Vanilla   |  83.01 |  83.01 |  82.35 |  81.70 |  79.08 |
> > > | Attack    |  82.35 |  82.35 |  81.17 |  79.74 |  78.43 |
> > > | Cost ($)  | 0.6948 | 0.6655 | 0.6381 | 0.5852 | 0.5273 |
> > >
> > > These results are consistent with the sensitivity trends already shown for $\rho$ and $\eta$ in **Fig. 3(c,d)**: stronger cost regularization lowers expense but also gradually reduces performance. We will correct this notation and wording in the revision.
> > >
> > > ---
> > > We thank the reviewer again for these helpful follow-up questions and for the careful reading, which has helped us identify and clarify these details more precisely.

---

### Decision · Program_Chairs · 2026-04-30

**Decision:**

Accept (regular)

**Comment:**

Reviewers appreciated that this work addresses an interesting, important, and timely problem, as well as the conceptual contribution of the proposed primitives and training signals. They further commended the thorough evaluation, extensive sensitivity study, and the promising empirical benefits. Through detailed discussion, the author addressed several reviewer concerns, including method clarity and the justification of some design choices (e.g. the reward expression), the optimization method, and the primitive vocabulary. One lingering concern is the limited evaluation of the extent to which the manual choice of primitives and the textual prompt-editing, rather than the trainable workflow generator, shape the outcome; the authors do show that the generator's optimization matters, but this doesn't fully resolve the question. If accepted, the authors are reminded to fix the presentation issues that reviewers identified.